# PM_2.5_ Concentrations Variability in North China Explored with a Multi-Scale Spatial Random Effect Model

**DOI:** 10.3390/ijerph191710811

**Published:** 2022-08-30

**Authors:** Hang Zhang, Yong Liu, Dongyang Yang, Guanpeng Dong

**Affiliations:** 1Key Research Institute of Yellow River Civilization and Sustainable Development, Henan University, Kaifeng 475001, China; 2Collaborative Innovation Center on Yellow River Civilization Jointly Built by Henan Province and Ministry of Education, Henan University, Kaifeng 475001, China; 3Key Laboratory of Geospatial Technology for the Middle and Lower Yellow River Regions, Ministry of Education, Kaifeng 475001, China

**Keywords:** spatial statistics, basis functions, heterogeneity, spatial correlation, PM_2.5_ concentrations

## Abstract

Compiling fine-resolution geospatial PM_2.5_ concentrations data is essential for precisely assessing the health risks of PM_2.5_ pollution exposure as well as for evaluating environmental policy effectiveness. In most previous studies, global and local spatial heterogeneity of PM_2.5_ is captured by the inclusion of multi-scale covariate effects, while the modelling of genuine *scale-dependent variabilities* pertaining to the spatial random process of PM_2.5_ has not yet been much studied. Consequently, this work proposed a multi-scale spatial random effect model (MSSREM), based a recently developed fixed-rank Kriging method, to capture both the *scale-dependent variabilities* and the spatial dependence effect simultaneously. Furthermore, a small-scale Monte Carlo simulation experiment was conducted to assess the performance of MSSREM against classic geospatial Kriging models. The key results indicated that when the multiple-scale property of local spatial variabilities were exhibited, the MSSREM had greater ability to recover local- or fine-scale variations hidden in a real spatial process. The methodology was applied to the PM_2.5_ concentrations modelling in North China, a region with the worst air quality in the country. The MSSREM provided high prediction accuracy, 0.917 R-squared, and 3.777 root mean square error (RMSE). In addition, the spatial correlations in PM_2.5_ concentrations were properly captured by the model as indicated by a statistically insignificant Moran’s *I* statistic (a value of 0.136 with *p*-value > 0.2). Overall, this study offers another spatial statistical model for investigating and predicting PM_2.5_ concentration, which would be beneficial for precise health risk assessment of PM_2.5_ pollution exposure.

## 1. Introduction

PM_2.5_ refers to particulate matters with an aerodynamic diameter ≤ 2.5 microns, which is not only a major lethal health factor in addition to hypertension, smoking, hyperglycaemia, and high cholesterol [1], but also causes great social and economic loss [2]. Precise health risk assessment of PM_2.5_ pollution exposure and environmental policy evaluation would require an accurate fine-resolution spatial data product and suitable modelling strategies [3,4]. However, this presents a major challenge.

From the formation mechanics perspective, PM_2.5_ takes the particles in the pollutant gas as condensation nuclei, with water vapour and other substances condensing on it, and thus, the pollutant gas emission (i.e., primary PM_2.5_) directly affects the PM_2.5_ concentrations [5]. In addition, the secondary PM_2.5_ formation process through complex photochemical reaction, condensation, and atmospheric processes tends to be highly variable across space and scales [6,7]. Thereby, a credible modelling approach is expected to capture such effects simultaneously and explicitly [8,9].

### 1.1. Classic Methods for Ground PM_2.5_ Concentrations

There are two types of methodologies commonly used to model and predict ground PM_2.5_ concentrations: the mechanistic approach and the statistical model approach. Mainstream mechanistic models, including the atmospheric transport model [10], community multiscale air quality [11], and the weather research and forecasting/chemistry [12], belong to a class of physical mechanics-driven digital simulation methods of pollutant concentrations. Despite their great ability in providing near real-time forecasting of PM_2.5_ concentrations at the global scale, such models are computationally intensive and often require computer clusters for implementation. This hinders their wide applications in applied environmental and social science research. It is also challenging to incorporate relatively accurate ground-monitoring sites-based measures of PM_2.5_ concentrations and potential socio-economic factors into the mechanistic models [13,14]. Moreover, uncertainties in the process of generating pollutant emission inventory data (e.g., the accuracy and timeline of emission inventories) and model implementation were hard to quantify [15,16]. 

Another mainstream approach to investigating ground PM_2.5_ concentrations and environmental variables is the spatial statistical model [17,18]. On one hand, this approach is flexible to cope with the linear or non-linear effects of potential factors of PM_2.5_ concentrations. On the other hand, it can model the spatial correlation and heterogeneous effects in the spatial distribution of PM_2.5_ concentrations. There appears to be a consensus that the spatial distribution of PM_2.5_ concentrations is significantly affected by both natural factors, such as elevation, landform, vegetation, and meteorological conditions [19,20], and human factors, such as population density, energy consumption, and economy [21,22]. The corresponding effects were treated as the determinate part (or global trends) in classic spatial statistical modelling [18]. Depending on the geographic scale where covariates are measured, the recent literature has tended to decompose the deterministic trend into a global component and a local component [23,24,25]. In addition, localised variabilities in the associations between covariates and PM_2.5_ concentrations, which are another important aspect of local variability, have also been modelled through a set of local spatial statistical approaches, such as geographically weighted regression models [26,27,28,29].

The most noteworthy features of the spatial statistical modelling approach lie in its rigorous and its explicit modelling of spatial correlations, which arises from the geographical proximity of locations [17,30]. Adding the spatially structured correlation effect into model specifications leads to at least two critical advantages. First, it produces valid and reliable statistical inferences on covariate effects [17,31], and thus offers a better approach compared to classic non-spatial statistical models for studies that seek to identify the potential significant factors. Secondly, with the spatial correlation structure constructed by random samples, spatial statistical models and various Kriging methods in particular lead to the best linear unbiased prediction for the spatial field [30,32]. Consequently, spatial or spatio-temporal statistical models have been widely applied to studies that scrutinize potential forces governing the PM_2.5_ concentrations spatial variabilities [33,34], and predict PM_2.5_ concentrations over a study area [35,36]. It is useful to note that various machine learning approaches have also been applied to produce national- and global-scale PM_2.5_ concentrations data products [37,38]; however, inherent spatial correlation structure and *scale-dependent variabilities* in the spatial random process beyond the deterministic trend of PM_2.5_ concentrations have not been explicitly modelled. 

### 1.2. Scale-Dependent Variabilitie, and Spatial Correlation in an Integrated Model

Most often, global and local spatial heterogeneity in the distributional surface of PM_2.5_ concentrations are modelled by the inclusion of multi-scale covariate effects [6,35], while the modelling of genuine *scale-dependent variabilities* pertaining to the spatial random process of PM_2.5_ concentrations has not yet been much studied. *Scale-dependent variabilities* can be understood as differential spatial patterns of PM_2.5_ concentrations (in general, an outcome variable of interest) observed from multiple scales. For instance, the distribution of PM_2.5_ concentrations might be smooth when viewed at an aggregated national or global scale but exhibits great discontinuities (or even abrupt changes) at a local or small scale. The co-existence of smoothness and discontinuities at different scales was highlighted as a generic feature of the distribution of geographical variables [39]. 

From a statistical modelling perspective, modelling *scale-dependent variabilities* and spatial correlations in a unified statistical model is challenging. In a seminal paper by Cressie and Johannesson (2008), an innovative method, the fixed-rank Kriging (FRK) model, was proposed [40]. FRK defines a spatially correlated mean-zero and generally nonstationary random process, which is further decomposed by using a linear combination of flexible and multi-scale spatial basis functions with structured random coefficients. By doing so, it can reconstruct a complex, spatially dependent, nonstationary, and high-dimensional spatial process. Moreover, this is scalable for large spatial datasets [18].

In line with the FRK model, we proposed a multi-scale spatial random effect model (MSSREM) to explore the spatial variability in ground PM_2.5_ concentrations in North China. This area was chosen because of its relatively high levels of air pollution and the great variabilities that it exhibits with regard to natural and socioeconomic characteristics. Before our empirical investigation, we first conducted a Monte Carlo simulation experiment to assess the relative prediction performance of the MSSREM against a single-scale spatial statistical model (e.g., a classical ordinary Kriging). The simulation results indicated that when higher levels of local spatial variabilities were exhibited, the MSSREM had a greater potential to recover local- or fine-scale variations hidden in spatial processes. Furthermore, we found significant impacts of both meteorological, physical, and human activity factors on the distribution of PM_2.5_ concentrations in North China.

The remainder of this paper is organized as follows: The statistical model is presented in Section 2. The description of a Monte Carlo simulation experiment is given in Section 3 to assess the relative prediction performance between MSSREM and single-scale spatial statistical models. In Section 4, we present our empirical study results. The conclusions are presented in Section 5.

## 2. Statistical Modelling

Our conceptual framework for PM_2.5_ spatial process modelling is presented in Figure 1. Briefly, we assume that the geographical process of PM_2.5_ concentrations is driven by regional factors, including nature and human factors, and a spatial random process. For a study region *R*, the hidden (or real) process of PM_2.5_, namely H(s), is defined as
(1)$$H\left(s\right)=N{\left(s\right)}^{T}\alpha +M{\left(s\right)}^{T}\beta +\omega \left(s\right)+\xi \left(s\right);\;s\in R,$$
where s denotes the location of H(s). On the right-hand side of the equation, the first two terms capture global deterministic trend of PM_2.5_ concentrations, in which $N{\left(s\right)}^{T}\alpha$ measures the effect of nature factors, and $M{\left(s\right)}^{T}\beta$ measures the contribution of human factors. The third term, ω(s), is spatial Gaussian process capturing the spatially structured random effect underlying the outcome variable. The last term, ξ(s), is a random error term with mean zero and variance-covariance $\sigma_{\xi }^{2}I$, which is spatially uncorrelated.

For the PM_2.5_ spatial process, in the real world, boundary effect and scale effect are unavoidable. Consequently, the spatial random process is decomposed as multi-scale spatial basis function with random coefficients [40],
(2)$$\tilde{\omega }\left(s\right)=\overset{r}{\underset{k=1}{\sum }}\Phi_{ks}\tau_{k}+\xi \left(s\right);\;s\in R,$$
where $\tau ={\left(\tau_{1},\ldots ,\tau_{r}\right)}^{T}$ is an *r*-dimensional Gaussian vector with mean zero and *r* by *r* covariance matrix **K**, and $\tau_{k}$ captures the average random effect governed by *k* th spatial basis function. $\Phi =\left(\Phi_{1},\ldots ,\Phi_{r}\right)$ is *r*-dimensional spatial basis functions (e.g., Gaussian basis function or exponential basis function) with a multi-scale nested structure (e.g., Figure 2). To cater for different observation supports (e.g., monitoring stations and remote sensing pixels), the region is discretized as n non-overlapping but compact, basic areal units (BAU) [41]. If BAUs are small enough, compared to the study region, the error in the discrete process could be ignored. Then, the hidden process, H(s), is averaged over the BAUs, which can be written as
(3)$$H\left(B_{i}\right)=\frac{1}{\left|B_{i}\right|}\int_{Bi}H\left(s\right)d\left(s\right);\;\;i=1,\;\ldots \;,\;n.$$
where $\left|B_{i}\right|$ is the area of BAU-i. At the BAU level, the process model can be written as
(4)$$H\left(B_{i}\right)=N{\left(B_{i}\right)}^{T}\alpha +M{\left(B_{i}\right)}^{T}\beta +\tilde{\omega }\left(B_{i}\right)+\xi \left(B_{i}\right);\;\;i=1,\;\ldots \;,\;n.$$

A simple illustration of the idea is provided in Figure 2. The region, R, is discretized as *n* BAUs, and spatial basis functions at three scales (different bandwidth in the kernel functions) are constructed to capture the heterogeneous random effects of PM_2.5_ concentrations. For the BAU with observations in Figure 2, such as BAU-1, its value is governed by the three spatial basis functions ($\Phi_{1}$, $\Phi_{2}$, and $\Phi_{3}$) and calculated as $\tilde{\omega_{1}}=0.4\times 0.55+0.6\times 0.25+0.3\times 0.2=0.43$. For the BAU without observations in Figure 2, such as BAU-0, its random effect is calculated as $\tilde{\omega_{0}}=0.4\times 0.6+0.6\times 0.23+0.3\times 0.17=0.429,$ with the same spatial basis functions ($\Phi_{1}$, $\Phi_{2}$ and $\Phi_{3}$) but with different weights. If a BAU was governed by a single spatial basis function, the variabilities on other scales would be ignored, such as $\tilde{\omega_{8}}=0.3\times 1=0.3$. Consequently, this multi-scale decomposition runs through the whole process of parameter estimation and prediction, leading to high flexibility to deal with complex variabilities and high computational efficiency.

When PM_2.5_ concentrations are measured either by monitoring stations or remote sensing instruments, measurement error is inevitable. Consequently, the measurement model is defined as the weighted average of hidden process plus an independent measurement error term,$\;\varepsilon_{j}$, as in Equation (1),
(5)$$P_{j}=\frac{\sum_{i=1}^{n}H\left(B_{i}\right)w_{ij}}{\sum_{i=1}^{n}w_{ij}}+\varepsilon_{j}\;\mathrm{and}\;w_{ij}=\frac{\left|O\left(P_{j}\right)\in B_{i}\right|}{O\left(P_{j}\right)};\;j=1,\ldots ,m,$$
where $O\left(P_{j}\right)$ denotes the footprint of observed PM_2.5_ concentration, $P_{j}$. $w_{ij}$ is the spatial weight between observation-i and BAU-j. For monitoring station data, $O\left(P_{j}\right)$ is the location of $P_{j}$, and $w_{ij}$ is a set of 0–1 weights. For remote sensing data, $O\left(P_{j}\right)$ is the area of $P_{j}$, and $w_{ij}$ is the overlapped area between pixel area-j and BAU-i. It is assumed that ***ε*** has a Gaussian distribution with mean-zero and variance-covariance $\sigma_{\varepsilon }^{2}I$. Here, $\sigma_{\varepsilon }^{2}$ is estimated using variogram techniques ahead of parameter estimation [42]. Eventually, if we define
(6)$$H_{j}=\frac{\sum_{i=1}^{n}H\left(B_{i}\right)w_{ij}}{\sum_{i=1}^{n}w_{ij}}\;,$$
the MSSREM can be written as
(7)$$P_{j}=N_{j}^{T}\alpha +M_{j}^{T}\beta +{\tilde{\omega }}_{j}+\xi_{j}+\varepsilon_{j};\;j=1,\;\ldots ,\;m.$$

The unknown parameters are included in a set $\vartheta =\left\{\alpha ,\beta ,\sigma_{\xi }^{2},K\right\}$. The MSSREM are estimated by the expectation-maximization (EM) algorithm. The complete-data likelihood is defined as L(ϑ)=[τ,P | ϑ]. After initialization, the EM algorithm for L(ϑ) is an iterative optimization procedure including E-step, which computes conditional distribution of τ based on Gaussian prior distribution at current parameter estimates (ϑ), and M-step, which updates ϑ based the conditional distribution of τ and finds the max-likelihood estimates.

## 3. A Monte Carlo Simulation Experiment

In this section, we conducted a small-scale Monte Carlo simulation study to assess the relative prediction performance between multi-scale spatial random effect model (MSSREM) and classic ordinary Kriging models (a single-scale spatial statistical model). The purpose was to demonstrate that MSSREM could serve as a useful methodology for modelling and predicting and to provide a tentative assessment on conditions under which MSSREM would be useful.

For simplicity, following Kang and Cressie (2011) and Sengupta and Cressie (2013), we chose a stable exponential spatial covariance function to generate a spatially correlated random field [43,44]:(8)$$C\left(d\right)=\sigma^{2}\exp\left(-{\left|d\right|}^{\alpha }\right);\alpha \in \left(0,2\right],$$
where C(d) is the covariance function related to distance ***d***; $\sigma^{2}$ is the variance of the field; and *α* is the power of distance. Under this specification, larger values of *α* indicate higher levels of stability or smoothness of spatial processes, as illustrated by Figure 3, where nine processes were generated with discrete values of *α* ranging from 0 to 2.

For a regular 200-by-200 grid topology with a resolution of 0.01°, 100 simulation experiments (random fields) were generated under each spatial covariance function scenario (i.e., 40 varied values of α with an equal interval of 0.05), leading to 4000 experiments for the 4000 grids on a two-dimensional lattice. We treated each simulated random field as a realisation (population in the statistics terminology) of the real PM_2.5_ concentrations process in region $R^{o}$, $H_{i}^{\alpha }:i\in \left[1,100\right]$. 

To assess the relative performance between MSSREM and the classic ordinary Kriging method, spatial point data and areal data commonly used in the studies of ground PM_2.5_ concentrations were chosen as experimental data. Kriging methods usually operate with point-level data, whereas MSSREM could process point-level data, areal data, or both at the same time. To mimic the real-world PM_2.5_ monitoring station data, under each simulation scenario (i.e., 40 varied values of *α* with an equal interval of 0.05), we randomly draw 500 points (grid centroids) from each simulated real random process as point-level sample data. With respect to areal sample data, we simply aggregated a real random process generated to a resolution of 0.1°. Two sample data are depicted in Figure 4.

For each of the 4000 experiments, the MSSREM and classic ordinary Kriging models were implemented, both running with an exponential spatial covariance function. Simple R-squared statistic was calculated to assess model fit (e.g., Cressie, 1993; Banerjee, Carlin and Gelfand, 2015). Results were presented in Figure 5. In line with common sense, when globally structured spatial variability (stronger spatial dependence) is exhibited, both methods could reasonably reconstruct the underlying real process with an acceptable error range, as indicated by high values of R-squared statistic (≥0.96) with values of *α* ≥ 1.5. This observation holds for both point and areal sample data.

When higher levels of local spatial variabilities exhibited, the MSSREM produced better model fit than the classic ordinary Kriging model did for both spatial point (α∈(0,0.675)) and areal data (α∈(0,0.525)), indicating that MSSREM had greater chances to recover local- or fine-scale variations hidden in spatial processes. When medium levels of local spatial variabilities exhibited, for instance, α∈(0.675,2) of point-level sample and α∈(0.525,2) of area-level sample, model fits produced by both methodologies were not really distinguishable. Overall, this small-scale simulation experiment suggested that the MSSREM model, due to the use of multi-scale spatial basis functions with random coefficients, performed relatively better than the classic ordinary Kriging model. This could present a real advantage of MSSREM in real-world empirical examinations of ground PM_2.5_ concentrations, where global or large-scale spatial variabilities were usually captured by covariate effects.

## 4. Empirical Study

### 4.1. Study Area, Data Sources, and Variables

#### 4.1.1. Study Area

North China is one of the five meteorological geographic zones, covering the regions of Beijing, Tianjin, Hebei, Shanxi, Shandong, and Henan. It sits to the north of the Qinling Mountains-Huaihe River line and the south of the Great Wall and has a significant topographic variability, being high in the West and low in the East (Figure 6). The region locates in the transition from subtropical to temperate zones, thus exhibiting great climatic differences between its north and south areas. Spatial disparities in socioeconomic and population distributions are also evident. The north region is one of areas with the worst air pollution levels in China and the world. Whether the combined differences in both natural and human factors lead to prominent variability in the PM_2.5_ concentrations, and if so, to what extent, are the key inquiries of our empirical study.

#### 4.1.2. Ground PM_2.5_ Concentrations

We crowded sourcing ground PM_2.5_ concentrations data by using web crawler technology (with python language) from the World Air Quality Project (http://aqicn.org (accessed on 10 July 2020)), a project providing historical and real-time air-quality data. To ensure model estimation robustness, we excluded stations with missing data for more than 65 days or 15 consecutive days and calculated annual ground PM_2.5_ concentrations averages for 1287 stations, as shown in Figure 6. The station data were part of the Nowcast system of The U.S. Environmental Protection Agency (EPA), which converted raw pollutant readings into air-quality index values (on a scale ranging from 0 to 500), referred to as the PM_2.5_ air quality index ($AQI_{pm_{2.5}}$) [45]. According to the US-EPA 2016 standard, we converted $AQI_{pm_{2.5}}$ back into PM_2.5_ concentrations ($C_{PM_{2.5}}$) based on the formula
(9)$$C_{PM_{2.5}}=\frac{\left(AQI_{PM_{2.5}}-AQI_{low}\right)}{\left(AQI_{high}-AQI_{low}\right)\left(C_{high}-C_{low}\right)}+C_{low},$$
where $C_{low}$ and $C_{high}$ are, respectively, the left and right boundaries of the subinterval that $C_{PM_{2.5}}$ falls into and belongs to the range with breakpoints (0, 12, 35, 55, 150, 250, 350, 500). $AQI_{low}$ and $AQI_{high}$ are, respectively, the breakpoints (0, 50, 100, 150, 200, 300, 400, 500) corresponding to $C_{low}$ and $C_{high}$.

#### 4.1.3. Independent Variables

Following Zhou et al. (2021) and Wei et al. (2020) [37,46] and the conceptual framework mentioned earlier, this study constructed nature and human factors to explain the deterministic trend in PM_2.5_ concentrations. Detailed sources and descriptions of covariates are presented in Table 1.

### 4.2. Empirical Model Specification

The empirical model specification follows Equation (7). Regular grids with a 0.02° × 0.02° resolution were chosen as the basic areal units, yielding 48,403 BAUs. To capture the potential *scale-dependent variabilities*, spatial basis functions at three scales (a large scale with 5.4° radius, a medium scale with 1.6° radius, and a small scale with 0.5° radius) were specified, as depicted in Figure 7. It is useful to note that there has not been a consensus on the optimal scale number of spatial basis functions [47]. However, in this study, the spatial basis functions with various spatial scales number were constructed, and the found model with a three-scale spatial basis function yielded the highest model fit.

### 4.3. Covariate Effects

Results on regression coefficients and the associated statistical significance of covariates are presented in Table 2. With respect to meteorological factors, relative humidity, cumulative precipitation, and wind speed were statistically negatively correlated with PM_2.5_ concentration, with everything else being equal. It is understandable that precipitation could clean the air by shooting down particles. Wind could accelerate PM_2.5_ escape speed, thus decreasing PM_2.5_ concentration, ceteris paribus. Higher temperature was associated with higher levels of PM_2.5_ concentration. In addition, the greenhouse effect of aerosols (PM_2.5_) could lead to warming [48], which could be a vicious cycle of air pollution and climate change in the study area and globally. 

With respect to land-use characteristics, only unused land density and woodland–grassland density were statistically negatively associated with PM_2.5_ concentration. In the human activity domain, there were no consistent evidences on significant relationships between industry concentration and PM_2.5_ concentration and between local urbanization and PM_2.5_ concentration, as indicated by the insignificant regression coefficients of covariates *IED* and *NTL*. The significant correlation between road network density and PM_2.5_ concentration might highlight the importance of transportation emission in air pollution.

### 4.4. Prediction Accuracy

This study used a tenfold cross-validation procedure to assess model fit and prediction accuracy. We randomly selected 90% of the data as the training group and the remaining 10% as validation group or out-of-sample validation. This whole procedure was repeated for 100 times, and results are presented in Figure 8. Following Cressie and Johannesson (2008) and Zammit-Mangion and Cressie (2021), the R-squared statistics and root mean squared error (RMSE) were used to assess prediction accuracy [40,47]. We noted that the MSSREM in Section 4.3 was fitted with full data, in which the validation group is the same as the training group. However, the sampling method of out-of-sample validation, a more robust verification method for predict accuracy in which the validation group is different from the training group, had a small probability to assign outliers into the validation group. This resulted in R^2^ in the full-data model being less than that in tenfold cross-validation.

As clearly presented in Figure 8a, the regression slope, obtained by regressing the predicted values on observed values of PM_2.5_ concentrations, was close to one on average, indicating a good model fit. In addition, the averaged R-squared value was as high as 0.917 with an interval of (0.914, 0.923) (mean ± 1.96 × standard error), whilst the averaged RMSE was 3.777 with an interval of (3.665, 3.889) (mean ± 1.96 × standard error). With respect to the spatial distribution of estimation errors, only 1.5% of the stations exhibited absolute estimation errors ≥15 and 75% of the stations with absolute estimation errors less than 5. More importantly, the distribution of model estimation errors appeared to be spatially random, which was confirmed by a statistically insignificant Moran’s *I* statistic of 0.136 (a *p*-value > 0.2). This highlighted that the spatial dependency effects were well-captured by the MSSREM model. 

Among existing studies, Wei et al. (2020) reconstructed the PM_2.5_ pattern in North China based on machine learning method and derived fitting results (R^2^ = 0.92 and RMSE = 11.52) [37]. Compared with this, our results with close R^2^ = 0.917 and evidently smaller RMSE = 3.777 show a higher precision. This is mainly because through the multi-scale local modelling of residual *scale-dependent variabilities* and spatial dependence effect outside the global trends, spatial basis functions with random coefficients well-recovered local variations hidden in spatial processes of secondary PM_2.5_ and ensured smaller local errors on a fine scale.

## 5. Conclusions

Producing high-accuracy and PM_2.5_ concentrations data at a fine spatial resolution is essential for health risk assessment and environment regulation evaluation. Primarily, PM_2.5_ concentrations is the key variable that links to various health outcome variables, and a fine spatial resolution pollution measure could yield a more accurate estimation of the relationships between pollution and health. This study presented a multi-scale spatial random effect model (MSSREM) for investigating PM_2.5_ concentrations’ variability. Besides the spatial correlation effects often observed for geographical data, it has the capacity to model the potential scale-dependent effect, as it is flexibly specified by a linear combination of multi-scale spatial basis functions. Beyond the conceptual modelling advantages, it substantively improves computational efficiency by estimating a much smaller set of spatial basis function coefficients rather than a full set of spatial random effects, thus offering great potential to cater for large spatial data. 

The small-scale simulation experiment indicates that when higher levels of local spatial variabilities are exhibited in a Gaussian random file, the MSSREM had greater chances to recover local- or fine-scale variations hidden in spatial processes, especially in real-world empirical examinations where global or large-scale spatial variabilities were usually captured by covariate effects. This was confirmed by the empirical study on North China based on MSSREM, in which we obtained more reliable covariate effects than non-spatial statistics and more precise prediction results with smaller local errors than previous studies.

In terms of methodological significance, the multi-scale modelling strategy developed in this study could, to some extent, alleviate the modifiable areal unit problem. As it captures the multiple-scale variabilities in the spatial random effect, the potential confounding effects between covariates and geographical scales could be substantially reduced. With respect to policy significance, compiling local- and fine-resolution PM_2.5_ concentrations data would be beneficial for precise health risk assessment of PM_2.5_ pollution exposure because a PM_2.5_ concentration data with smaller local errors offer opportunities to understand the nuanced relationships between air pollution and health. In addition, with medium effects, it is intuitive to extend our methodology to a spatio-temporal modelling context, thus offering a practical solution to obtain fine spatio-temporal-scale PM_2.5_ concentration estimates, contributing real-time monitoring of regional air pollution. 

Despite a careful design for investigating the annual PM_2.5_ concentrations variability in the North China, some limitations remain. Firstly, remote-sensing-based data were not simultaneously modelled along with the monitoring station data although the multi-scale spatial random effect model, in principle, can model multiple data sources with different spatial supports. Secondly, the annual average left the temporal variabilities unmodelled. However, a further methodological extension to a simultaneously modelling monitoring station and remote sensing-based PM_2.5_ concentrations data as well as the temporal dependency is on top of our future research priorities. 

## Figures and Tables

**Figure 1 ijerph-19-10811-f001:**
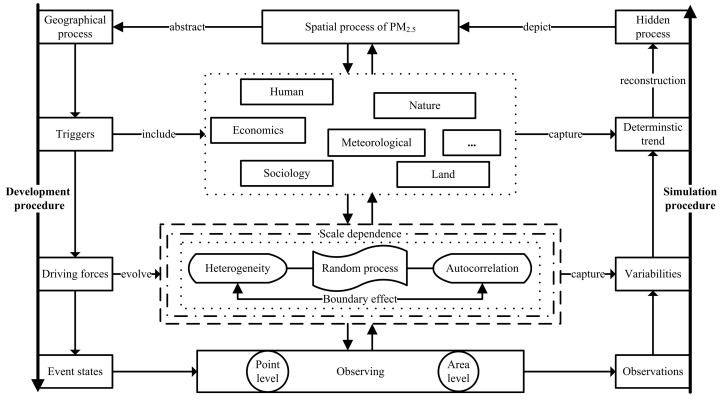
Modelling framework of PM_2.5_ spatial process.

**Figure 2 ijerph-19-10811-f002:**
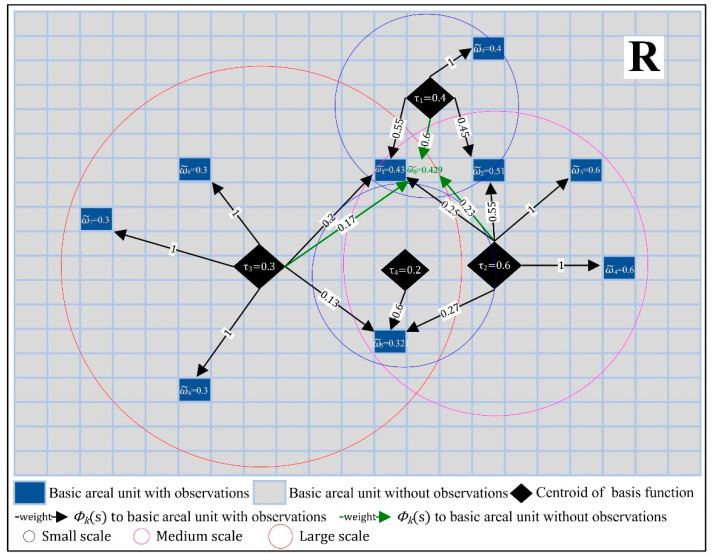
An illustration of heterogeneous random process captured by multi-scale spatial basis function.

**Figure 3 ijerph-19-10811-f003:**
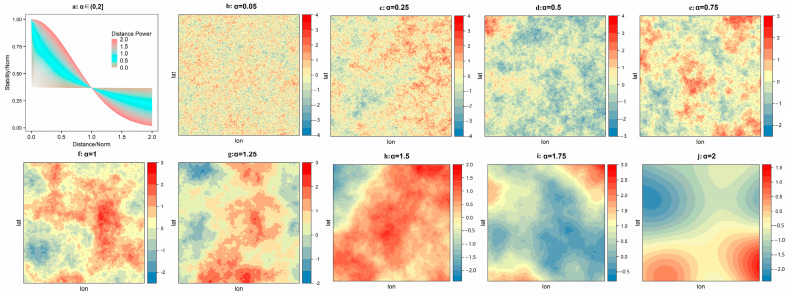
Simulated Gaussian random fields under an exponential spatial covariance function with different values of *α*.

**Figure 4 ijerph-19-10811-f004:**
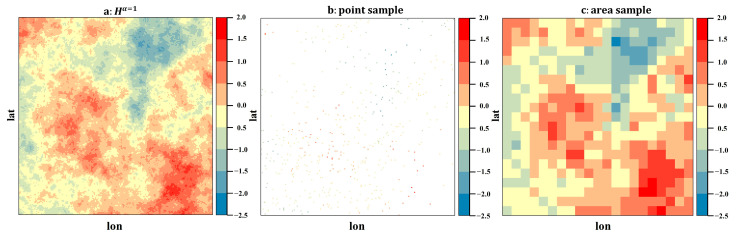
Real process and sampling data in the case of *α* = 1.

**Figure 5 ijerph-19-10811-f005:**
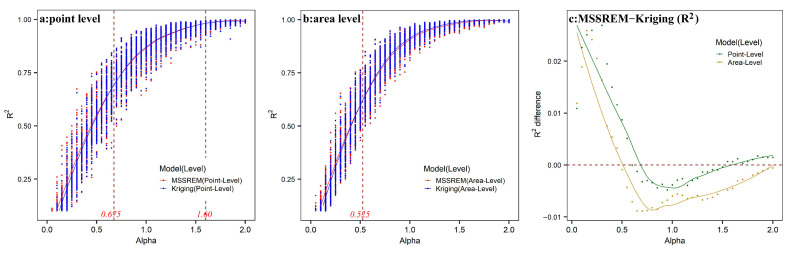
(**a**) Point-level modelling accuracy in MSSREM and ordinary Kriging; (**b**) area-level modelling accuracy in MSSREM and ordinary Kriging; (**c**) difference in accuracy between MSSREM and ordinary Kriging (
$R_{\mathrm{MSSREM}}^{2}-R_{\mathrm{oK}}^{2}$).

**Figure 6 ijerph-19-10811-f006:**
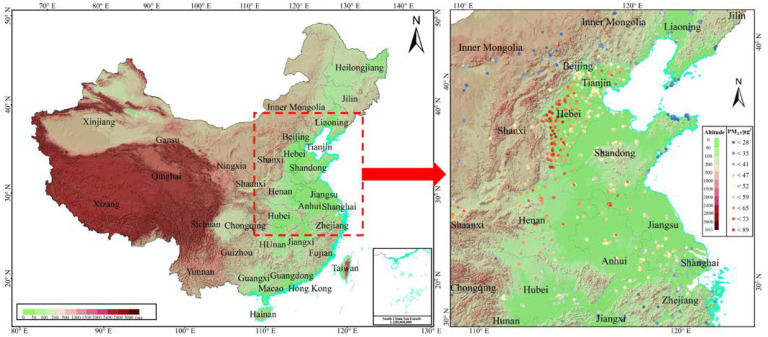
The geographical distribution and topographic features of North China.

**Figure 7 ijerph-19-10811-f007:**
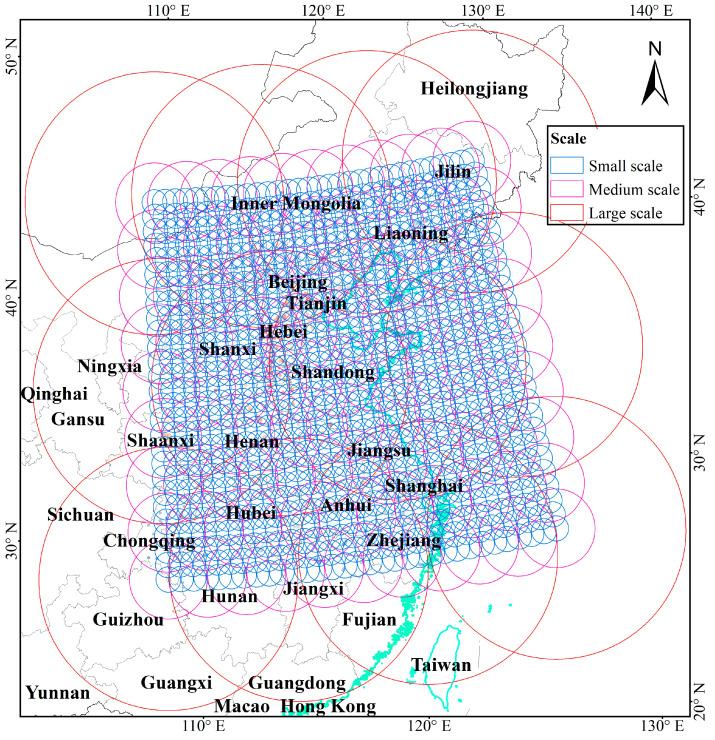
The scope of Gaussian spatial basis functions with three scales.

**Figure 8 ijerph-19-10811-f008:**
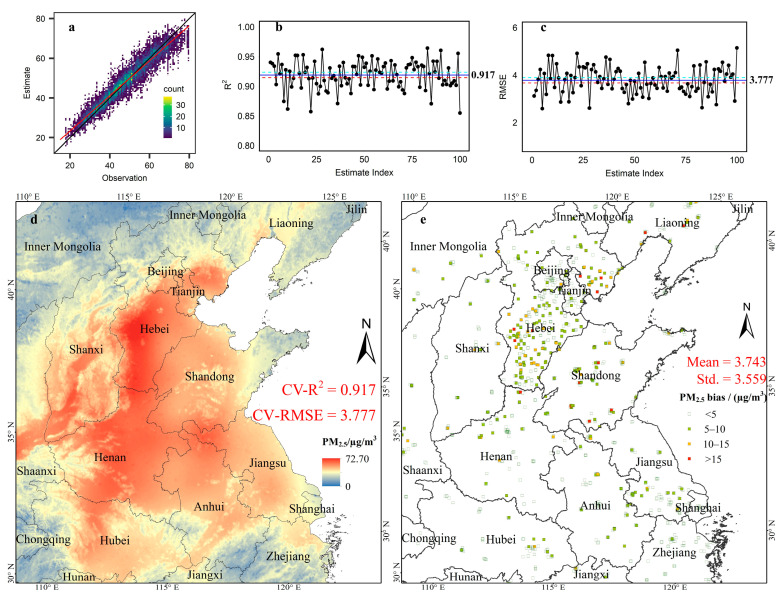
(**a**) Scatter density plot of observations and estimations; (**b**) scatter diagram of R-squared; (**c**) scatter diagram of root mean square errors (RMSE); (**d**) prediction of PM_2.5_ concentrations in North China; (**e**) spatial distribution of estimation errors.

**Table 1 ijerph-19-10811-t001:** Description of the data sources used in the study.

Data Domain	Variable	Content	Unit	Spatial Resolution	Data Source	Computing Method
PM_2.5_	*P*	Particulate Matter ≤ 2.5 µm	µg m^−3^	In Situ	AQICN	Denoising
Meteorology	*TEM*	2 m air temperature	K	0.1° × 0.1°	CMA	Interpolation
	*RLH*	Relative humidity	%	0.1° × 0.1°	CMA	Interpolation
	*CPP*	Cumulative precipitation	Mm	0.1° × 0.1°	CMA	Interpolation
	*WDS*	10 m wind speed	m s^−1^	0.1° × 0.1°	CMA	Interpolation
Land use	*WGD*	Woodland–grassland density	%	0.1° × 0.1°	CNLUCC	Kernel Density
	*CSD*	Construction land density	%	0.1° × 0.1°	CNLUCC	Kernel Density
	*UUD*	Unused land density	%	0.1° × 0.1°	CNLUCC	Kernel Density
	*CTD*	Cultivated land density	%	0.1° × 0.1°	CNLUCC	Kernel Density
Altitude	*DEM*	DEM	M	0.1° × 0.1°	SRTM-V4.1	Denoising
Human activity	*IED*	Industry–enterprise density	%	0.1° × 0.1°	Amap	Kernel Density
	*RND*	Road network density	%	0.1° × 0.1°	Amap	Quadrat Sample
	*NTL*	Night-time lights	W cm^−2^ sr^−1^	0.1° × 0.1°	NPP-VIIRS	Denoising

Notes: CMA refers to China Meteorological Administration; CNLUCC refers to China land use and land cover change origin from Resource and Environmental Science and Data Centre, Chinese Academy of Sciences; SRTM refers to American Shuttle Radar Topography Mission.

**Table 2 ijerph-19-10811-t002:** Model estimation results from MSSREM.

DataDomain	Variables	Coefficients	Standard Error	*t*-Value *	*p*-Value
Meteorology	*TEM*	0.287	0.011	26.534	0.000
*RLH*	−0.411	0.046	8.846	0.000
*CPP*	−1.947	0.069	28.159	0.000
*WDS*	−0.988	0.056	17.497	0.000
Landuse	*WGD*	−10.840	1.854	5.846	0.000
*CSD*	−0.709	1.667	0.425	0.671
*UUD*	−25.117	10.037	2.502	0.012
*CTD*	−2.698	1.711	1.577	0.115
Altitude	*DEM*	−0.010	0.001	17.001	0.000
Humanactivity	*IED*	−2.800	2.532	1.106	0.269
*RND*	0.013	0.006	2.288	0.022
*NTL*	−0.004	0.012	0.338	0.736
Others	*Intercept*	85.617	3.057	28.004	0.000
R^2^	0.855
RMSE	5.137

## Data Availability

The air-quality data are available at https://aqicn.org/data-platform/register/ (accessed on 10 July 2020). The Meteorology data are available at https://data.cma.cn/ (accessed on 12 September 2020). The land-use and altitude data are available at https://www.resdc.cn/ (accessed on 22 August 2020). The industry–enterprise location and road network data are sourced from Amap-API (https://lbs.amap.com/, accessed on 15 October 2018). The night-time lights data are available at https://eogdata.mines.edu/products/vnl/ (accessed on 17 October 2020).

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
