# Peer review of "PM2.5 Concentrations Variability in North China Explored with a Multi-Scale Spatial Random Effect Model"

_ijerph, 2022, doi:10.3390/ijerph191710811_

Round 1
Reviewer 1 Report
The article is important for the explored PM2.5 concentration variability and appropriate to the journal theme. The results are important to explore the health risks of PM2.5 pollution exposure. In conclusion, a further methodological extension to simultaneously modelling monitoring station and remote sensing-based PM2.5 concentration data, as well as the temporal dependency, is on top of our future research priorities ok. And environmental policies?? (suggestion, discuss the importance).

Author Response
Please see the attachment, in which the point-by-point response to the two reviewers' comments are provided.

Reviewer 2 Report
Dear Authors,
The study's subject falls into the target Journal's aims.
As a first consideration, it is necessary to indicate that air pollution modelling is a relevant aspect of public health to protect human beings' health. In this sense, particulate matter modelling is paramount, especially in particles with an aerodynamic diameter lower than 2.5 µm. The structure of the paper follows the scientific directrices.
General comments
The editing of English should be improved; there are grammatical errors in the paper.
Specific comments
In the abstract appear acronyms that have not previously been defined.
The objective of publishing a research work is the dissemination of its scientific content and potential use by other researchers. For this reason, the authors should upgrade the methodological development, which should be enhanced to ease its implementation by other research groups worldwide (major revision). In this sense, the methodology reported is complex to understand.
Figure 6. The text could be highlighted to visualize better.
Author Response

(The authors gave the same response as above.)
